# Two C_23_-Steroids and a New Isocoumarin Metabolite from Mangrove Sediment-Derived Fungus *Penicillium* sp. SCSIO 41429

**DOI:** 10.3390/md22090393

**Published:** 2024-08-30

**Authors:** Lishan Huang, Chunmei Chen, Jian Cai, Yixin Chen, Yongyan Zhu, Bin Yang, Xuefeng Zhou, Yonghong Liu, Huaming Tao

**Affiliations:** 1Guangdong Provincial Key Laboratory of Chinese Medicine Pharmaceutics, School of Traditional Chinese Medicine, Southern Medical University, Guangzhou 510515, China; huanglishan0527@163.com (L.H.); 13676126834@163.com (Y.C.); yongyanzhu0521@163.com (Y.Z.); 2CAS Key Laboratory of Tropical Marine Bio-Resources and Ecology, Guangdong Key Laboratory of Marine Materia Medica, South China Sea Institute of Oceanology, Chinese Academy of Sciences, Guangzhou 510301, China; chenchunmei18@mails.ucas.ac.cn (C.C.); caijian19@mails.ucas.ac.cn (J.C.); yangbin@scsio.ac.cn (B.Y.); xfzhou@scsio.ac.cn (X.Z.); yonghongliu@scsio.ac.cn (Y.L.)

**Keywords:** C_23_-steroid, isocoumarin, *Penicillium*, mangrove-sediment fungus, antioxidant, pancreatic lipase (PL)

## Abstract

Two new C_23_-steroids derivatives, cyclocitrinoic acid A (**1**) and cyclocitrinoic acid B (**2**), and a new isocoumarin metabolite, (3*R*,4*S*)-6,8-dihydroxy-3,4,5-trimethyl-7-carboxamidelisocoumarin (**10**), together with 12 known compounds (**3**–**9**, **11**–**15**) were isolated from the mangrove-sediment fungus *Penicillium* sp. SCSIO 41429. The structures of the new compounds were comprehensively characterized by 1D and 2D NMR, HRESIMS and ECD calculation. All isolates were evaluated for pancreatic lipase (PL) inhibitory and antioxidant activities. The biological evaluation results revealed that compounds **2**, **14** and **15** displayed weak or moderate inhibition against PL, with IC_50_ values of 32.77, 5.15 and 2.42 µM, respectively. In addition, compounds **7**, **12** and **13** showed radical scavenging activities against DPPH, with IC_50_ values of 64.70, 48.13, and 75.54 µM, respectively. In addition, molecular docking results indicated that these compounds had potential for PL inhibitory and antioxidant activities, which provided screening candidates for antioxidants and a reduction in obesity.

## 1. Introduction

In recent decades, the prevalence of obesity has demonstrated a significant global escalation and has become a major public health challenge that needs to be addressed [1]. In this context, the biological function of pancreatic lipase (PL), a key enzyme in the regulation of fat metabolism, has received increasing attention from the scientific community. PL, which is mainly synthesized by human pancreatic cells and secreted into the duodenum, plays a central role in the hydrolysis of dietary fats, and is an important target for the control of fat absorption and metabolism [2]. Currently, PL inhibitors, such as orlistat, have become commonly used in the clinical treatment of obesity to reduce fat absorption by effectively inhibiting PL activity, thereby achieving weight loss. However, the widespread use of these drugs is also accompanied by a series of side effects, especially gastrointestinal discomfort, which limits the safety of their long-term use and patient compliance [3]. Therefore, it is important to explore and develop efficient and safe novel weight loss drugs to replace or optimize the existing therapeutic regimens for the management of obesity and its related metabolic diseases.

A variety of human diseases occur due to the excessive production of reactive oxygen species (ROS) in the body during oxidative stress states, capable of attacking and damaging biomolecules [4]. Moreover, antioxidants, as a class of compounds capable of slowing down or blocking oxidative processes, show great potential in the prevention and treatment of oxidative stress-related diseases. They protect cells from damage by providing electrons or hydrogen atoms to oxidative substrates and neutralizing ROS [5]. The wide application of antioxidants is not only limited to the pharmaceutical field but is also commonly found in food preservation and cosmetic industries, as exemplified by using ascorbic acid (vitamin C) in the preservation of fruits and vegetables. Because of the above background, this study aimed to screen potential antioxidant candidate molecules using the DPPH radical scavenging assay as a rapid and effective means of assessing antioxidant activity, which provides a scientific basis for subsequent in-depth research and application development.

Marine fungi have proven to be a very promising source of bioactive compounds [6,7,8]. In particular, mangrove fungi, a unique and biodiversity-rich group of marine fungi, have demonstrated a strong secondary metabolite production capacity, providing a wealth of candidate molecules for drug discovery [9]. *Penicillium* spp. fungi are common microbial taxa in mangrove ecosystems. From 2007 to 2020, 276 new secondary metabolites isolated and identified from mangrove *Penicillium* spp. of mangrove origin were reported, of which 140 compounds exhibited biological activity [10]. During our continuous study exploring new bioactive natural products from mangrove fungi [11,12,13,14], two unusual C_23_-steroids (Appendix A), cyclocitrinoic acid A (**1**) and cyclocitrinoic acid B (**2**), one new isocoumarin derivative, (3*R*,4*S*)-6,8-dihydroxy-3,4,5-trimethyl-7-carboxamidelisocoumarin (**10**), and 12 known natural products (**3**–**9**, **11**–**15**) (Figure 1) were obtained from the fungus *Penicillium* sp. SCSIO 41429, which was isolated from mangrove sediment. Herein, we described the fermentation, isolation, structural determination and biological activities of these compounds.

## 2. Results and Discussion

### Structural Identification of New Compounds

Compound **1** was obtained as a yellow solid, and its molecular formula was established as C_23_H_30_O_4_ by HERESIMS ion peak at *m*/*z* 369.2081 [M − H]^−^ (calcd for C_23_H_29_O_4_^−^, 369.2071), indicating nine degrees of unsaturation. Its IR spectrum exhibited absorption bands at 3392.79 and 1672.28 cm^−1^, indicative of hydroxy and carbonyl groups, respectively. The ^1^H NMR showed signals for three alkene hydrogens at *δ*_H_ 5.66 (1H, s, H-22), 5.56 (1H, dd, *J* = 8.5, 6.0 Hz, H-1), and 5.43 (1H, s, H-7), and two methyls at *δ*_H_ 2.10 (3H, s, H_3_-21) and 0.53 (3H, s, H_3_-19). The ^13^C NMR and DE PT spectra (Table 1) revealed 23 carbon-atom resonances including one carbonyl at *δ*_C_ 204.1 (C-6); one carboxyl at *δ*_C_ 167.6 (C-23); four quaternary carbons at *δ*_C_ 156.5 (C-20), 156.5 (C-8), 145.4 (C-10), and 47.1 (C-13); eight methine carbons at *δ*_C_ 48.1 (CH-5), 53.1 (CH-9), 54.5 (CH-14), 59.9 (CH-17), 63.0 (CH-3), 117.4 (CH-22), 122.2 (CH-1), and 124.5 (CH-7); seven methylene carbons at *δ*_C_ 22.3 (CH_2_-16), 23.9 (CH_2_-15), 27.2 (CH_2_-18), 27.4 (CH_2_-11), 35.9 (CH_2_-2), 36.9 (CH_2_-12), and 41.4 (CH_2_-4); and two methyl carbons at *δ*_C_ 13.3 (CH_3_-19) and 19.8 (CH_3_-21). The details of 1D-NMR data and 2D-NMR spectra (Figure 2) showed that **1** has a similar tetracyclic steroid skeleton with a bicyclo [4.4.1] A/B ring system to the reported compound, neocyclocitrinol A [15]. On the side chain, compound **1** and neocyclocitrinol A both exhibited signals for a methyl carbon at *δ*_C_ 19.8 (CH_3_-21), and double-bond carbons at *δ*_C_ 156.5 (C-20) and 117.4 (CH-22). However, they differed in that compound **1** lacked signals for C-24 and C-25, and exhibited a carboxylate signal at *δ*_C_ 167.6 (C-23) instead of the tertiary alcohol carbon. Moreover, the fragment of C20–C23 of compound **1** was supported by the HMBC correlations (Figure 2) of 2.42 (H-17)/156.5 (C-20), 2.10 (H_3_-21)/156.5 (C-20), 5.66 (H-22)/59.9 (CH-17), 5.66 (H-22)/156.5 (C-20) and 5.66 (H-22)/167.6 (C-23).

The NOEs between pairs of the protons (Figure 2), H-3/H-9, H-9/H-17, H-17/Hα-12 and Hα-12/H-14, indicated that orientations of H-3, H-9, H-14, and H-17 were on the same side of the A–D ring system, and H-17/Hα-15, H_3_-19/Hβ-15, H-5/H_3_-19, indicated that H-5 and H_3_-19 were on the other side of the ring system in **1**, so the relative configuration of **1** was *Rel*-(3*S*, 5*S*, 9*R*, 13*R*, 14*R*, 17*R*). Then, the ECD calculation (Figure 3) showed that the absolute configuration of compound **1** was 3*S*, 5*S*, 9*R*, 13*R*, 14*R*, 17*R*. Finally, the structure of **1** was determined as shown in Figure 1 and named cyclocitrinoic acid A.

Compound **2** was obtained as white solid, and its molecular formula was established as C_23_H_32_O_5_ by HERESIMS ion peak at *m*/*z* 389.2320 [M + H]^+^ (calcd for C_23_H_33_O_5_^+^, 389.2323), indicating eight degrees of unsaturation. Its IR spectrum exhibited absorptions bands at 3392.79 and 1697.36 cm^−1^, indicative of hydroxy and carbonyl groups, respectively. Comparison of the ^1^H and ^13^C NMR (Table 1) data revealed that **2** was an analogue of **1**. The obvious differences were the changed chemical shifts at C-20 (−84.5 ppm), CH_2_-22 (−69.5 ppm), CH_3_-21 (+7.6 ppm), and C-23 (+6.0 ppm) and the replacement of a methine group (*δ*_H_ 5.66 (H-22)/*δ*_C_ 117.4 (CH-22) in **1** by a methylene group (*δ*_H_ 2.24 (H_2_-22)/*δ*_C_ 72.0 (CH_2_-22)) in **2**. Thus, it was implied that compound **2** differed in the absence of double-bond Δ^20,22^ and the hydroxylation at C-20. The above deduction was supported by the HMBC correlations of 1.64~1.80 (H-17)/72.0 (C-20), 1.30 (H_3_-21)/72.0 (C-20), 2.24 (H_2_-22)/72.0 (C-20), and 2.24 (H_2_-22)/173.5 (C-23).

The NOEs between pairs of the protons (Figure 2), H-5/H_3_-19, indicated orientations of H-5 and H_3_-19 on the same side of the A–D ring system, and H-9/H-14, H-9/Hα-12, H-14/H-17, Hα-12/H-21, Hα-11/H-14, and Hβ-11/H_3_-19, indicated H-9, H-14, H-17 and H-21 on the other side of the ring system in **2**. It was postulated that the A–D rings of compounds **1** and **2** were in the same absolute configurations because they were analogues that had similar ECD curves, despite the absence of NOESY signals of H-3. It was possible to confirm that the absolute configuration of C-3 in compound **2** is 3*S*. To determine the absolute configuration of the chiral center C-20, the ^13^C NMR chemical shift calculation of two possible stereoisomers, (3*S*, 5*S*, 9*R*, 13*S*, 14*R*, 17*S*, 20*R**)-**2** and (3*S*, 5*S*, 9*R*, 13*S*, 14*R*, 17*S*, 20*S**)-**2**, was performed using the gauge including atomic orbital (GIAO) method. The calculation results showed that the experimental NMR data for **2** most closely match those of (3*S*, 5*S*, 9*R*, 13*S*, 14*R*, 17*S*, 20*R*)-**2** based on DP4+ probability analysis of carbon chemical shift (Figure 4), suggesting the absolute configuration of **2** was established as 3*S*, 5*S*, 9*R*, 13*S*, 14*R*, 17*S*, 20*R* and named as cyclocitrinoic acid B.

Compound **10** was obtained as yellow solid, and its molecular formula was established as C_13_H_15_NO_5_ by HRESIMS ion peak at *m*/*z* 266.1025 [M + H]^+^ (calcd for C_13_H_16_NO_5_, 266.1023). The unsaturation degree of seven and IR spectrum exhibited bands at 3402.16 cm^−1^ (hydroxyl group) and 1638.37 cm^−1^ (carbonyl group). A detailed analysis of ^1^H NMR data (Table 2) of **10** exhibited the presence of two methines at *δ*_H_ 4.86 (q, *J* = 6.6 Hz, H-3) and 3.18 (q, *J* = 7.1 Hz, H-4); and three methyls at *δ*_H_ 2.01 (s, H_3_-13), 1.22 (d, *J* = 6.6 Hz, H_3_-12), and 1.17 (d, *J* = 7.1 Hz, H_3_-11). The ^13^C NMR data and HSQC spectrum displayed 14 carbon signals including two ester carbons at *δ*_C_ 175.1 (C-7-CONH_2_) and 170.2 (C-1); six olefinic tertiary carbons at *δ*_C_ 168.8 (C-6), 166.0 (C-8), 144.8 (C-10), 111.9 (C-5), 100.4 (C-7), and 94.20 (C-9); two methine carbons at *δ*_C_ 80.2 (CH-3) and 33.9 (CH-4); and three methyl carbons at *δ*_C_ 18.6 (CH_3_-11, 12) and 9.1 (CH_3_-13). The ^1^H-^1^H COSY correlations of H_3_-11/H-3 and H_3_-12/H-4 revealed partial structures of CH-3/CH_3_-11 and CH-4/CH_3_-12. The above NMR data indicated that **10** had the same isocoumarin skeleton as the co-isolated **11**. The main distinction was the presence of an amide group at C-7 of **10** instead of the alkene hydrogen at C-7 of **11**, which was supported by HSQC correlations of 14.86 (H_2_-7-CONH_2_)/175.1 (C-7-CONH_2_) and 100.4 (C-7) of **10** instead of 6.29 (H-7)/100.2 (C-7). Thus, the planar structure of **10** was defined as shown in Figure 5, and the other HMBC correlations supported the deduction. To determine the absolute configurations of C-3/C-4, the NOESY analysis and ECD calculation methods were used. The NOESY correlations of H-3/H_3_-12 and H-4/H_3_-11 supported the different orientations of CH_3_-11 and CH_3_-12. As shown in Figure 5, the experimental ECD was matched to the calculated ECD spectrum of (3*R*,4*S*)-**10**. The ^1^H coupling constant data of H-3 (4.86, q, *J* = 6.6 Hz) and H-4 (3.18, q, *J* = 7.1 Hz) of compound **10** were similar to H-3 (4.68, q, *J* = 6.6 Hz) and H-4 (3.06, q, *J* = 7.1 Hz) of the known compound **11**. The data provided further evidence of the differing orientations of CH_3_-11 and CH_3_-12. Consequently, the absolute configuration of **10** was determined as 3*R*,4*S* and named (3*R*,4*S*)-6,8-dihydroxy-3,4,5-trimethyl-7-carboxamidelisocoumarin.

In addition to the isolation of the above new compounds **1**, **2**, and **10**, **12** known compounds were isolated, including *cyclo*-(l-Pro-l-Tyr) (**3**) [16], *cyclo*-(l-Phe-l-Ala) (**4**) [17], guinolactacin A1 (**5**) [18], *N*-(*N*-acetyl-valyl)-phenylalanine (**6**) [19], butyrolactone I (**7**) [20], penicillenol A1 (**8**) [21], penicillenol A2 (**9**) [21], stoloniferol B (**11**) [22], decarboxydihydrocitrinin (**12**) [23], phenol A (**13**) [24], 4-hydroxy-3,5,6-trimethyl-2H-pyran-2-one (**14**) [25], and 4-methyl-5,6-dihydropyren-2-one (**15**) [26]. These structures were elucidated through a comparison of their NMR and MS data with reported literature.

All compounds (**1**–**15**) were evaluated for pancreatic lipase inhibitory and antioxidant activity in vitro, according to the reported methods [27,28,29]. Compounds **2**, **14**, and **15** displayed weak or moderate inhibition against PL, with IC_50_ values of 32.77, 5.15, and 2.42 µM, respectively. Orlistat was used as a positive control, with IC_50_ value of 0.079 µM. Meanwhile, compounds **7**, **12**, and **13** showed radical scavenging activities against DPPH, with IC_50_ values of 64.70, 48.13, and 75.54 µM, respectively. Ascorbic acid was used as a positive control, with a IC_50_ value of 42.61 µM.

To further understand the interaction between the compounds and proteins, docking studies were carried out for compounds in the active site of PL (PDB ID: 1ETH) and superoxide dismutase (PDB ID: 7wx0) to gain insights into their molecular interactions.

As a result, these ligands were favorably accommodated within the binding cleft with analogous anchoring conformations. Compounds **2** and **14** exhibited the binding free energy of −5.84 and −5.23 kcal/mol. Their grid box size was 78 × 58 × 126, centered at x: 72.007, y: 31.282, z: 145.92. As shown in Figure 6, Compound **2** formed one hydrogen-bonding interaction with residue ARG-65, and four hydrophobic-bonding interactions with residues GLU-64, LEU-357, TYR-370 and GLU-371, and displayed two salt bridges with residue LYS-42 and ARG-65. Compound **14** revealed one hydrogen bond with residue SER-78, and five hydrophobic-bonding interactions with residues PHE-78, TYR-115, PRO-181 and PHE-216, and exhibited a salt bridge with residue with HIS-264, and a π-stacking with residue PHE-78. The results of this molecular docking study of compound **15** were by those reported previously [30]. The result as reported showed that compound **15** exhibited the binding free energy of −5.82 kcal/mol, and the dimensions of the grid box size used was 78 × 58 × 126, centered at x: 72.007, y: 31.282, z: 145.92. Compound **15** interacted with residues ASN-329 and AGR-340 via two hydrogen bonds, and three hydrophobic bonds were formed with residues ALA-282, PHE-284 and ARG-340. Compounds **2**, **14** and **15** could inhibit PL by tightly binding to catalytic amino acid residues through diverse types of interactions.

Compound **2** was a derivative of compound 1 after hydrolysis of the double-bond at C-20 and C-22, but compound **1** did not show biological activity, suggesting that the hydroxyl group at C-20 of compound **2** was the key for PL inhibitory activity. Compounds **14** and **15** showed similar PL inhibitory activities, which suggested that the lactone group may be the key acting group. Compounds **10** and **11** were both isocoumarins. Despite the presence of lactone groups, they did not exhibit PL inhibitory activity. It was postulated that the benzene ring adjacent to the lactone group may prevent their interaction with PL enzyme active sites, thereby resulting in their inactivity.

Furthermore, molecular docking between the active compounds **7**, **12**, and **13** with superoxide dismutase (PDB ID: 7wx0) was performed to gain functional and structural insight (Figure 6). The results showed that compounds **7**, **12**, and **13** displayed the binding free energy of −6.06, −5.59, and −5.49 kcal/mol, respectively. The size of compounds **7**, **12**, and **13** was 60 × 58 × 96, centered at x: 14.313, y: 6.955, z: 67,949. Compound **7** interacted with the residues ASN-86, THR-88, ILE-96 and GLU-100 via five hydrogen bonds, and a salt bridge interaction was formed with residue LYS-75, and displayed three hydrophobic interactions with residues with PRO-74 and ILE-99. Compound **12** revealed five hydrogen-bonding interactions with residues VAL-8, LYS-10, CYS-146 and VAL-148, and one hydrophobic-bonding interaction with residue VAL-8. Compound **13** formed six hydrogen-bonding interactions with residues VAL-8, LYS-10, CYS-146 and VAL-148, and four hydrophobic interactions with residues LYS-10, ASN-53 and VAL-148. The docking studies suggested that compounds **7**, **12**, and **13** could inhibit superoxide dismutase by tightly binding to catalytic amino acid residues through diverse types of interactions.

## 3. Materials and Methods

### 3.1. General Experimental Procedures

Optical rotations were measured in MeOH using a PerkinElmer MPC 500 (Hertford, UK). IR spectra were recorded on an IR Affinity-1 (Shimadzu, Beijing, China) and Bruker Tensor II spectrometer (Ettlingen, Germany). UV spectra were acquired on a UV-2600 spectrophotometer (Shimadzu, Kyoto, Japan). NMR spectra were obtained on a Bruker AVANCE spectrometer (500 and 700 MHz for ^1^H NMR; 125 and 175 MHz for ^13^C NMR (Broker, Fallanden, Switzerland)) using TMS as an internal standard. HRESIMS spectra were generated on a Bruker miXis TOF-QII mass spectrometer (Bruker, Fallanden, Switzerland). ECD data were measured by use of a Chirascan circular dichroism spectrometer (Applied Photophysics, Surrey, UK). Semi-preparative HPLC was performed on the Hitachi Primide using an ODS column (YMC-pack ODS-A (Kyoto, Japan), 10 × 250 mm, 5 µm).

### 3.2. Fungal Material

The fungus *Penicillium* sp. SCSIO 41429 used for this study was isolated from mangrove sediment collected in Zhangjiang, Guangdong province, China (21.235° N, 110.451° E). Currently, the strain was stored on MB agar (malt extract of 15 g, sea salt of 10 g, agar of 16 g, H_2_O of 1 L, and pH 7.4–7.8) slants at 4 °C and deposited at the Key Laboratory of Tropical Marine Bio-resources and Ecology, Chinese Academy of Sciences. The strain was identified as a *Penicillium* sp. (GenBank No. PP940097) by the ITS sequence of its rDNA. Thus, the strain was identified as *Penicillium* sp. SCSIO 41429.

### 3.3. Fermentation, Extraction, and Isolation

The fungus *Penicillium* sp. SCSIO 41429 was cultivated on the plate of MB agar for 5 days. The mycelia of the strain were cut into small pieces (1 × 1 × 0.5 cm^3^) and inoculated into 110 × 1000 mL Erlenmeyer flasks each containing Sabouraud’s dextrose broth (3 g of peptone, 12 g of glucose, 0.03 g of chloramphenicol and 1 L of water, pH = 5.6 ± 0.2) for 45 days at 25 °C. The fermented whole broth (33 L) was filtered through cheesecloth to separate into filtrate and mycelia. The filtrate was extracted three times with EtOAc, while mycelia were extracted four times with CH_3_OH. The EtOAc and CH_3_OH solutions were concentrated under reduced pressure to get dark brown gum, respectively, and then were combined to obtain crude extract (30 g).

The crude extract was subjected to silica gel CC using step gradient elution with petroleum ether/EtOAc (0–100%, *v*/*v*) and EtOAc/CH_3_OH (0–100%, *v*/*v*) to obtain four subfractions (Frs.1–4) based on TLC properties.

Fr.2 was subjected by MPLC with ODS column (MeOH/H_2_O, 0% to 100% MeOH) to obtain 11 subfractions (Fr.2.1–Fr.2.11). Fr.2.2 was separated by semipreparative reverse-phase HPLC (38% MeOH/H_2_O, 3 mL/min) to afford **14** (6.6 mg, *t*_R_ = 19.1 min). Fr.2.4 was purified by semipreparative reverse-phase HPLC (28%MeOH/H_2_O, 3 mL/min) to yield **13** (12.9 mg, *t*_R_ = 11.5 min), **10** (6.3 mg, *t*_R_ = 18.5 min), and **12** (16.6 mg, *t*_R_ = 32.5 min). Fr.2.7 was purified by semipreparative reverse-phase HPLC (61% MeOH/H_2_O, 3 mL/min) to give **1** (4.2 mg, *t*_R_ = 9.5 min), **7** (8.6 mg, *t*_R_ = 11.5 min), **8** (13.8 mg, *t*_R_ = 17.8 min), and **9** (13.3 mg, *t*_R_ = 19.5 min). Fr.3 was subjected by further MPLC with ODS column (MeOH/H_2_O, 0% to 100% MeOH) to obtain 10 subfractions (Fr.3.1–Fr.3.10). Fr.3.2 was purified by reverse-phase HPLC (20% MeOH/H_2_O, 3 mL/min) to give **4** (2.6 mg, *t*_R_ = 10.5 min), **5** (14.7 mg, *t*_R_ = 12.5 min) and **3** (14.7 mg, *t*_R_ = 17.6 min). Fr.3.10 was purified by reverse-phase HPLC (53% MeOH/H_2_O, 3 mL/min) to give **6** (10.3 mg, *t*_R_ = 20.0 min), **2** (3.3 mg, *t*_R_ = 21.5 min), and **11** (2.0 mg, *t*_R_ = 24.0 min). Fr.4 was separated by MPLC with ODS column (MeOH/H_2_O, 0% to 100% MeOH) to obtain six subfractions (Fr.4.1–Fr.4.6). Fr.4.6 was purified by semipreparative reverse-phase HPLC (8% MeCN/H_2_O, 3 mL/min) to yield **15** (4.5 mg, *t*_R_ = 30.5 min).

### 3.4. Structural Elucidation of the New Compounds ***1***, ***2***, and ***10***

Cyclocitrinoic acid A (**1**): white solid; ${\left[\alpha \right]}_{D}^{25}$ = + 89.5°, (c 0.1, MeOH); UV (MeOH) *λ*_max_ (log *ε*) 240 (0.9) nm; IR (film) *v*_max_ 3392.79, 1672.28, 1644,71, 1298.09, 1246.02, 1174.65, 1033.85, 867.97 cm^−1^; HRESIMS at *m*/*z* 369.2081 [M − H]^−^ (calcd for C_23_H_29_O_4_^−^, 369.2071); ECD (0.25 mg/mL, MeOH) *λ*_max_ (Δ*ε*) 225 (17.05), 251 (11.20), 319 (−9.22) nm. ^1^H and ^13^C NMR: see Table 1.

Cyclocitrinoic acid B (**2**): yellow solid; ${\left[\alpha \right]}_{D}^{25}$ = +95.2°, (c 0.1, MeOH); UV (MeOH) *λ*_max_ (log *ε*) 245 (0.5) nm; IR (film) *v*_max_ 3392.79, 1697.36, 1672,21, 1456.26, 1396.46, 1205.51, 1024.20, 896.90, 829.39 cm^−1^; HRESIMS at *m*/*z* 389.2320 [M + H]^+^ (calcd for C_23_H_33_O_5_^+^, 389.2323); ECD (0.25 mg/mL, MeOH) *λ*_max_ (Δ*ε*) 218 (3.13), 246 (19.26), 318 (−9.00) nm. ^1^H and ^13^C NMR: see Table 1.

(3*R*,4*S*)-6,8-Dihydroxy-3,4,5-trimethyl-7-carboxamidelisocoumarin (**10**): yellow solid; ${\left[\alpha \right]}_{D}^{25}$ = +23.9°, (c 0.1, MeOH); UV (MeOH) *λ*_max_ (log *ε*) 240 (1.90), 290 (0.98), 340 (0.88) nm; IR (film) *v*_max_ 3402.16, 3212.06, 1638.37 cm^−1^; HRESIMS at *m*/*z* 266.1025 [M + H]^+^ (calcd for C_13_H_16_NO_5_^+^, 266.1023); ECD (0.25 mg/mL, MeOH) *λ*_max_ (Δ*ε*) 212 (11.32), 232 (12.53), 257 (4.44), 290 (14.00) nm. ^1^H and ^13^C NMR: see Table 2.

### 3.5. Pancreatic Lipase Inhibition Activity In Vitro Assay

The in vitro PL inhibitory activity of compounds **1**–**15** was evaluated with the established protocol [27,28]. A 2.5 mg/mL solution of pancreatic lipase was prepared by dissolving it in Tris–HCl buffer (pH = 8.4). The test sample was mixed with enzyme buffer and incubated at 37 °C for 10 min. Subsequently, *p*-nitrophenyl palmitate was added, and the enzyme reaction was permitted to continue at 37 °C for a further 10 min. PL activity was ascertained through measurement of the hydrolysis of *p*-nitro-phenyl palmitate to *p*-nitrophenol at 405 nm in a microplate reader. All compounds were subjected to a primary screening for enzyme inhibition activity at a concentration of 50 µg/mL, with orlistat as the positive control.

### 3.6. DPPH Free Radical Scavenging Assay

2,2-Diphenyl-1-picrylhydrazyl (DPPH) assay was employed to assess the free radical scavenging activities of all compounds. A specific experimental procedure was employed, based on the methodology proposed by Yen and Chen [29]. Then, 100 µL of 0.2 mM DPPH prepared in MeOH was mixed with 100 µL of each test sample ranging from 10 to 100 µg/mL in 96-well plates. After incubation for 30 min at room temperature in the dark, the absorbance of the resulting solution was measured spectrophotometrically at 517 nm. Ascorbic acid was used as the positive control, with MeOH serving as the blank. Decreased absorbance in the reaction mixture indicates heightened free radical scavenging efficacy. The scavenging ability of the compounds was calculated using the standard equation. IC_50_ values were calculated using the Origin 2018 software via a non-linear curve-fitting approach.

### 3.7. Molecular Docking Analysis

The molecular docking simulation was implemented by utilizing software AutoDock tools (ADT 1.5.6). The crystal structures of PL (PDB ID: 1ETH) and superoxide dismutase (PDB ID: 7wx0) were obtained from the Protein DataBank. The structures of ligands were generated in ChemBioOffice 18.0 (ChemBioOffice version 14.0), followed by an MM2 calculation to minimize the conformation energy. The other docking parameters, settings, and calculations were defaulted, and the docking results were analyzed using the software PyMOL 2.4.0.

### 3.8. ECD Calculations

Conformational searches were conducted by Spartan’14 software with the Merck molecular force field (MMFF). The stable conformers were subsequently optimized using Gaussian09 software at the B3LYP/6-31G(d) level in the gas phase. The optimized stable conformers were selected for further ECD calculations at the B3LYP/6-311G(d,p) level in methanol. The overall ECD data were weighted by Boltzmann distribution with a half-bandwidth of 0.3 eV using GuassView6.0 software, and ECD curves were produced by Origin 2018 after UV correction. Meanwhile, the best optimized conformer of 2 was chosen for NMR chemical shift calculations performed by GIAO at the PCM/mPW1PW91/6-311+G(d,p) level in dimethylsulfoxide.

## 4. Conclusions

Two C_23_-steroids, cyclocitrinoic acid A (**1**) and cyclocitrinoic B acid (**2**), and a new isocoumarin metabolite, (3*R*,4*S*)-6,8-dihydroxy-3,4,5-trimethyl-7-carboxamidelisocoumarin (**10**), together with 12 known metabolites were isolated from the mangrove-sediment fungus *Penicillium* sp. SCSIO 41429, which had been fermented using Sabouraud’s dextrose broth. The new structures, including absolute configurations, were identified through the application of spectroscopic methods coupled with the calculated ECD. During the screening for pancreatic lipase inhibitory and antioxidant activity, compounds **2**, **14**, and **15** displayed weak or moderate inhibition against PL, with IC_50_ values of 32.77, 5.15, and 2.42 µM, respectively. It was suggested that these compounds might be of further research value as potential pancreatic lipase inhibitor candidates in the management of obesity and the treatment of related metabolic diseases. Meanwhile, compounds **7**, **12**, and **13** showed radical scavenging activities against DPPH, with IC_50_ values of 64.70, 48.13, and 75.54 µM, respectively, which suggested their potential application as antioxidants in healthy food or drug development. The present study not only augmented the chemical diversity of secondary metabolites of mangrove fungi but also illuminated the prospective applications of some of these compounds in the domain of PL inhibitory and antioxidant activity through bioactivity screening. This provided a valuable material foundation and scientific basis for subsequent drug discovery and development studies.

## Figures and Tables

**Figure 1 marinedrugs-22-00393-f001:**
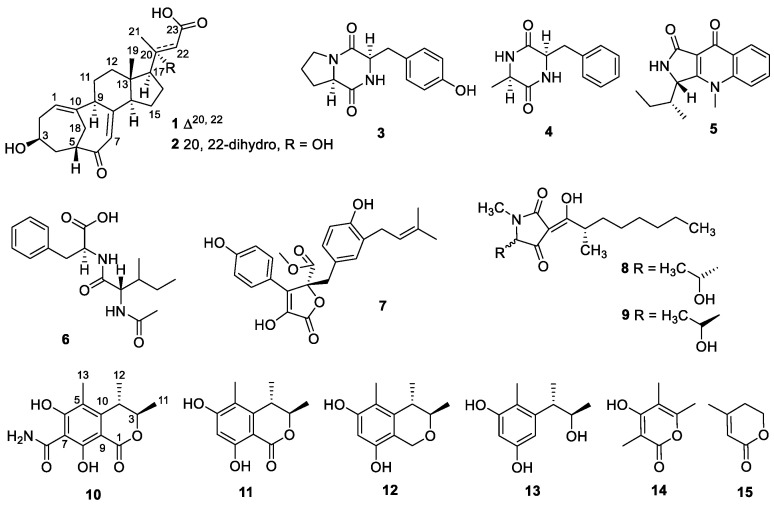
Structures of compounds **1**–**15**.

**Figure 2 marinedrugs-22-00393-f002:**
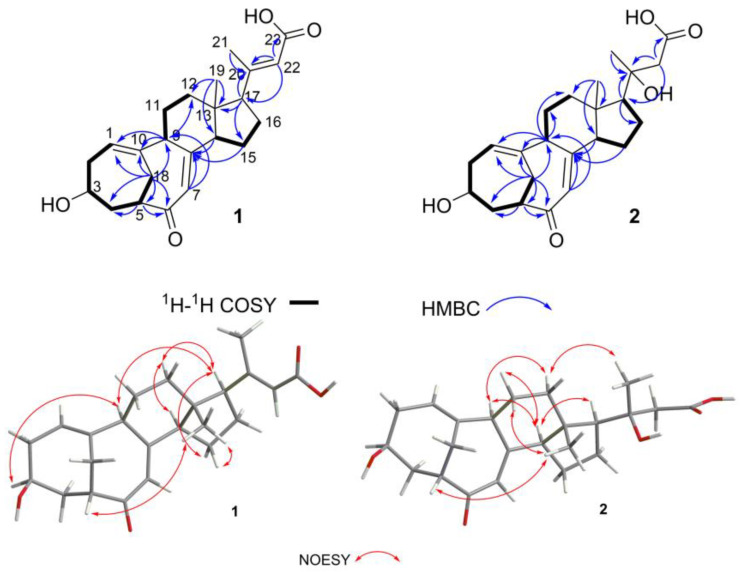
The key ^1^H-^1^H COSY, HMBC, and NOESY correlations of **1** and **2**.

**Figure 3 marinedrugs-22-00393-f003:**
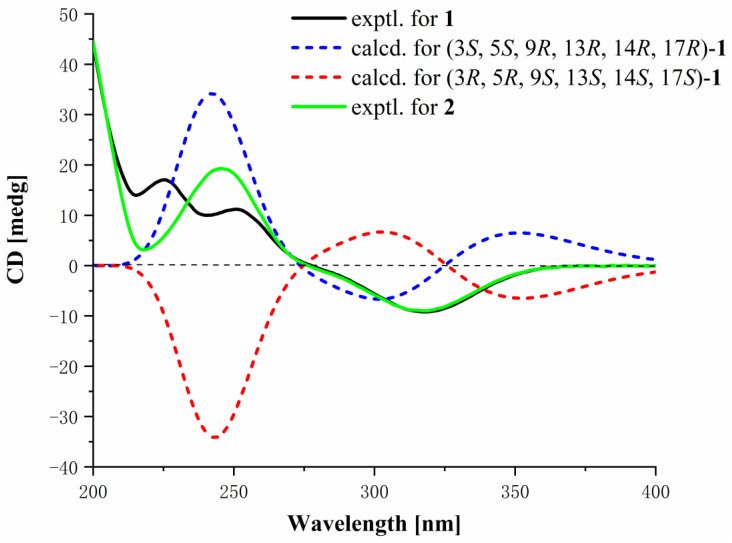
The experimental and calculated ECD spectra of **1**, and experimental of ECD spectra of **2**.

**Figure 4 marinedrugs-22-00393-f004:**
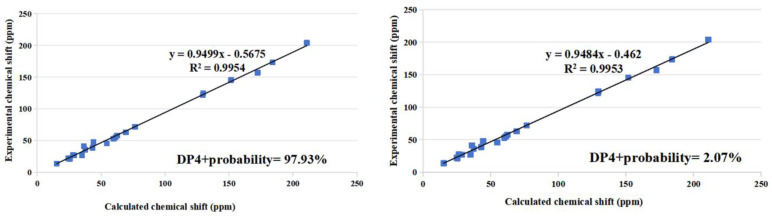
Linear regression analysis of calculated ^13^C NMR shifts of (3*S*, 5*S*, 9*R*, 13*S*, 14*R*, 17*S*, 20*R*)-**2** (**left**) and (3*S*, 5*S*, 9*R*, 13*S*, 14*R*, 17*S*, 20*S*)-**2** (**right**) against the experimental shifts of **2** and the DP4+ probability for assignment of **2** to the candidate stereoisomers.

**Figure 5 marinedrugs-22-00393-f005:**
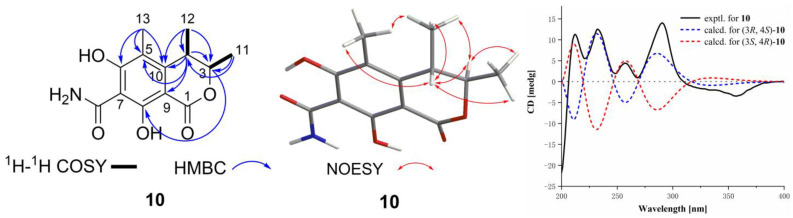
The key ^1^H-^1^H COSY, HMBC, and NOESY correlations of **10** and experimental and calculated ECD spectra of **10**.

**Figure 6 marinedrugs-22-00393-f006:**
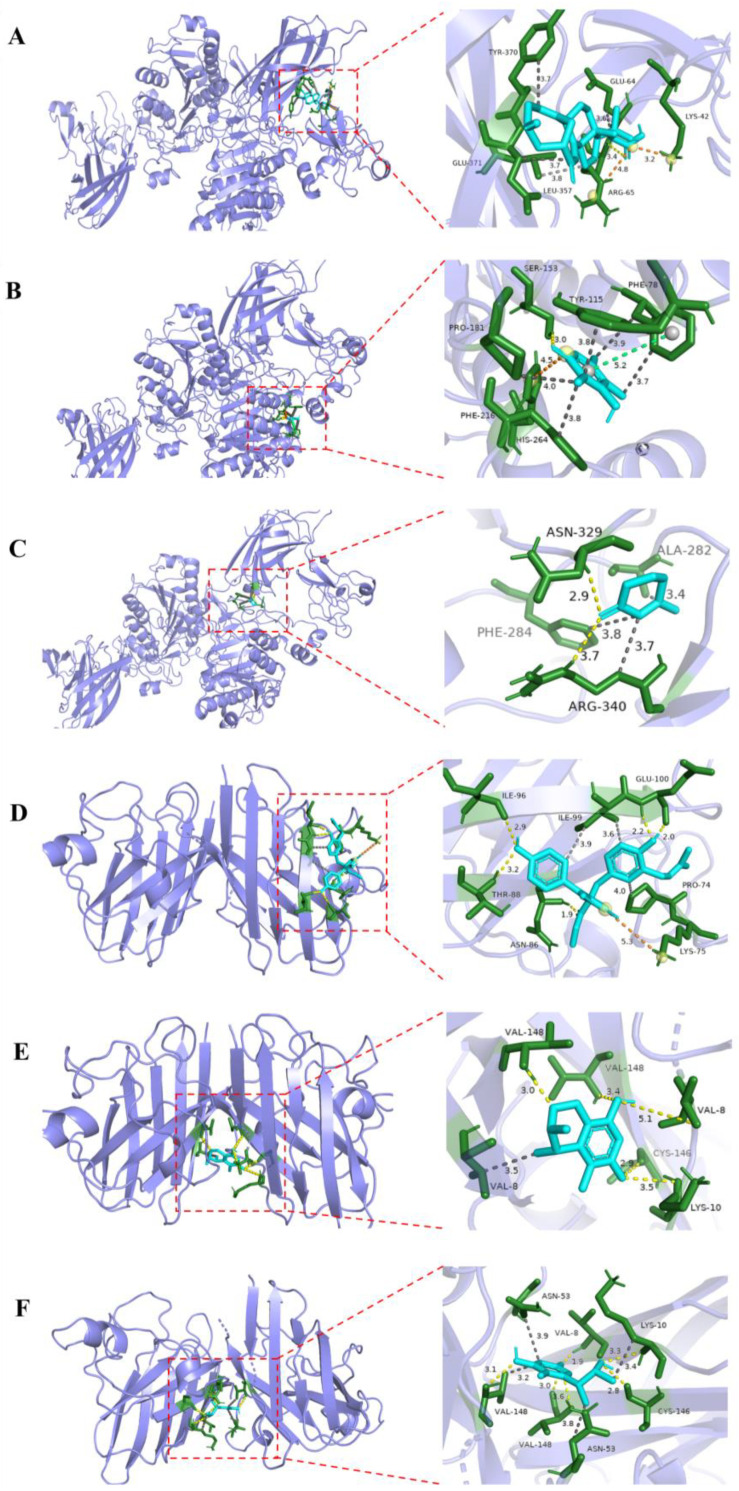
Molecular docking proposed binding interaction of compounds **2** (**A**), **14** (**B**) and **15** (**C**) with the active site residues of PL (PDB ID: 1ETH), and compounds **7** (**D**), **12** (**E**), and **13** (**F**) with the active site residues of superoxide dismutase (PDB ID: 7wx0). Yellow dotted line: hydrogen bond; gray dotted line: hydrophobic interaction; orange dotted line: salt bridge; green dotted line: *π*–*π* stacking interaction.

**Table 1 marinedrugs-22-00393-t001:** The ^1^H (500 MHz) and ^13^C (125 MHz) NMR data of **1** and **2** in DMSO-*d*_6_.

Pos.	1		2	
	*δ*_C_, Type	*δ*_H_ (*J* in Hz)	*δ*_C_, Type	*δ*_H_ (*J* in Hz)
1	122.2, CH	5.56 (dd, 8.5, 6.0)	122.0, CH	5.53 (dd, 8.5, 6.0)
2α	35.9, CH_2_	2.04~2.08 (overlapped)	35.9, CH_2_	2.04~2.13 (overlapped)
2β		2.28~2.37 (overlapped)		2.34 (m)
3	63.0, CH	3.12 (m)	63.0, CH	3.12 (m)
4α	41.4, CH_2_	1.50~1.62 (overlapped)	41.3, CH_2_	1.41~1.53 (overlapped)
4β		2.64 (m)		2.63 (m)
5	48.1, CH	2.69 (m)	48.1, CH	2.68 (m)
6	204.1, C		204.1, C	
7	124.5, CH	5.43 (s)	124.6, CH	5.39 (s)
8	156.5, C		156.9, C	
9	53.1, CH	2.86 (dd, 12.0, 5.5)	53.1, CH	2.81 (dd, 12.0, 5.5)
10	145.4, C		145.6, C	
11α	27.4, CH_2_	1.50~1.62 (overlapped)	27.2, CH_2_	1.41~1.53 (overlapped)
11β		1.71~1.80 (overlapped)		1.64~1.80 (overlapped)
12α	36.9, CH_2_	1.71~1.80 (overlapped)	38.8, CH_2_	2.04~2.13 (overlapped)
12β		1.50~1.62 (overlapped)		1.41~1.53 (overlapped)
13	47.1, C		45.7, C	
14	54.5, CH	2.28~2.37 (overlapped)	55.3, CH	2.04~2.13 (overlapped)
15α	23.9, CH_2_	1.71~1.80 (overlapped)	22.1, CH_2_	1.41~1.53 (overlapped)
15β		1.85 (m)		1.41~1.53 (overlapped)
16α	22.29, CH_2_	1.50~1.62 (overlapped)	21.6, CH_2_	1.64~1.80 (overlapped)
16β		1.50~1.62 (overlapped)		1.87 (m)
17	59.9, CH	2.42 (t, 9.5)	58.0, CH	1.64~1.80 (overlapped)
18	27.2, CH_2_	2.48 (overlapped)	27.1, CH_2_	2.47 (overlapped)
19	13.3, CH_3_	0.53 (s)	13.8, CH_3_	0.76 (s)
20	156.5, C		72.0, C	
21	19.8, CH_3_	2.10 (s)	27.4, CH_3_	1.30 (s)
22	117.4, CH	5.66 (s)	47.9, CH_2_	2.24 (d, 5.0)
23	167.6, C		173.5, C	

**Table 2 marinedrugs-22-00393-t002:** The ^1^H (500 MHz) and ^13^C (125 MHz) NMR data of **10** in DMSO-*d*_6_.

Pos.	*δ*_C_, Type	*δ*_H_ (*J* in Hz)
1	170.2, C	
3	80.2, CH	4.86 (q, 6.6)
4	33.9, CH	3.18 (q, 7.1)
5	111.9, C	
6	168.8, C	
7	100.4, C	
8	166.0, C	
9	94.2, C	
10	144.8, C	
11	18.6, CH_3_	1.22 (d, 6.6)
12	18.6, CH_3_	1.17 (d, 7.1)
13	9.1, CH_3_	2.01 (s)
7-CONH_2_	175.1, C	14.86 (s)

## Data Availability

The data presented in this study are available on request from the corresponding author.

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
