# Peer review of "Two C23-Steroids and a New Isocoumarin Metabolite from Mangrove Sediment-Derived Fungus Penicillium sp. SCSIO 41429"

_marinedrugs, 2024, doi:10.3390/md22090393_

Round 1

Reviewer 1 Report (Previous Reviewer 2)

Comments and Suggestions for Authors

I have reviewed the revised manuscript along with the authors' comments and corrections in response to the reviewers' concerns. I conclude that the manuscript has been revised and corrected according to the requested comments and is now suitable for publication.

Author Response

Thank you sincerely for your careful examination.

Reviewer 2 Report (New Reviewer)

Comments and Suggestions for Authors

Finding new types of pancreatic lipase inhibitors is a critical endeavor for the treatment of obesity. Tao and co-workers reported the chemical and biological investigation of the mangrove sediment-derived Fungus Penicillium sp. As a result, two new steroids with an uncommon carbon framework were obtained, along with a new isocoumarin and twelve known compouds. Interestingly, three compounds exhibited moderate inhibitory activity against pancreatic lipase. In addition, three compounds displayed antioxidant acitivity. Based on these findings, this work is suggested to be published in the forthcoming issue of this journal.

However, revisions were required as followings:

1. For compound 1, according to Figure S9, the IR absorption band 1679.28 cm-1 should be revised as 1672.28 cm-1. And 3392.79 cm-1 was not annotated in Figure S9. Similarly, for compound 2, the 1679.36 cm-1 should be revised as 1697.36 cm-1, and 3392.79 cm-1 was not annotated in Figure S20.

2. P3L85: It is better to describe more specifically as ‘four quaternary carbons’ instead of ‘four carbons’.

3. For compound 1, since H-17 was β-orientated and H3-19 was on α-facial, it is better to redefine Hα-15 and Hβ-15, and revise the expression as ‘H-17α/H -15, H3-19/Hβ-15’(P3L101). Similar for Hβ-11/H-14 for compound 2.

4. The obvious difference between compounds 2 and 1 was not clearly described. Indeed, they differed at the absence of double bond Δ20,22 accompanied with the hydroxylation at C-20. This was indicated by the remarkable upfield-shifted chemical shifts of C-20 and C-22, and led to the downfield-shifted chemical shifts of C-21 and C-23.

5. For compound 3, the correlations of 14.86 (H2-7-CONH2)/175.1 (C-155 7-CONH2) and 100.4 (C-7) were not observed neither in the HSQC spectrum (Figure S25) nor in the HMBC spectrum (Figure S26). The substituted position of the CONH2 group should be clearly supported by other HMBC correlations.

6. The expression ‘the trans-configuration’ was confusing. It is better to revise it as ‘the different orientations of CH3-11 and CH3-12’. And check the confusing expression ‘the known trans-compound 11’, too.

7. What was the criteria to judge that the experimental ECD was matched to the calculated ECD spectrum of (3R,4S)-10? As displayed in Figure 5, both calculated ECD curves partially match the experimental one.

8. Compounds 10 and 11 possess the lactone group but they did not show PL inhibitory activity. The SAR for compounds 14 and 15 should be re-analyzed.

Others:

1. P3: As described in the Experimental Section of ref. [15], the name of compound 9 was neocyclocitrinol A, not neocyclocitrinols A.

2. P3L105: showed in (Figure 1)showed in Figure 1’. And check the following typo errors ‘showed in (Figure 5)’(P6L157& L160).

Comments on the Quality of English Language

There were a few typo or grammar errors, some of which were given in the comments.

Round 2

Reviewer 2 Report (New Reviewer)

Comments and Suggestions for Authors

The authors well addressed my comments. And there were only minor revisions to be done.

1. Please add the answer ‘Compounds 10 and 11 were both isocoumarins. Despite the presence of lactone groups, they didn’t exhibit PL inhibitory activity. It was postulated that the benzene ring adjacent to the lactone functional group may prevent their interaction with PL enzyme active sites, thereby resulting in their inactivity’ in the manuscript.

2. Please check peak pattern of H2-22 of compound 2. It was unlikely ‘m’, because there was no proton on the adjacent carbons.

3. The pages for references  [24] and [25] were still missing.

Author Response

The authors well addressed my comments. And there were only minor revisions to be done.

Comments 1: Please add the answer ‘Compounds 10 and 11 were both isocoumarins. Despite the presence of lactone groups, they didn’t exhibit PL inhibitory activity. It was postulated that the benzene ring adjacent to the lactone functional group may prevent their interaction with PL enzyme active sites, thereby resulting in their inactivity’ in the manuscript.

Reponse 1: We added it. Thank you!

Comment 2:  Please check peak pattern of H2-22 of compound 2. It was unlikely ‘m’, because there was no proton on the adjacent carbons.

Reponse 2:  We have revised peak pattern of H2-22 as '2.24 (d, 5.0)' in Table 1 in the revised manuscript. Thank you.

Comment 3: The pages for references  [24] and [25] were still missing.

Reponse 3: Thank you for your suggestion. We rechecked carefully and revised it.

This manuscript is a resubmission of an earlier submission. The following is a list of the peer review reports and author responses from that submission.

Round 1

Reviewer 1 Report

Comments and Suggestions for Authors

The manuscript described the isolation and structure elucidation of natural products from a Mangrove Sediment Derived Fungus Penicillium sp. SCSIO 41429. All isolates were evaluated for pancreatic lipase (PL) and antioxidant inhibition activities. The biological evaluation results revealed compounds 2, 14, and 15 display weak or moderate inhibition against PL, with IC50 value of 32.77, 5.15 and 2.42 μM, respectively. In addition, compounds 7, 12, and 13 showed radical scavenging activities against DPPH, with IC50 values of 64.7, 48.13, and 75.54 μM, respectively. In addition, molecular docking results indicate that these compounds have potential for pancreatic enzyme inhibition and antioxidant activity. The results are of kind interest. Some minor modifications are needed before its acceptance for publication.

(1) In Figures 2 and 5, the structures should be numbered.

(2) In Figure 5, the HMBC signals should be checked and revised.

(3) It is interesting that the diversity of structures in this fungus is so high including steroids, alkaloids, isocoumarins etc., the other compounds have analogues but there are few derivatives of compound 7.

(4) In Figures, images should be combined.

(5) "Reference" should be re-arranged according to the Guidelines of the Journal.

(6) The language in “Introduction” still needs optimization.

Comments on the Quality of English Language

Minor editing of English language required

Reviewer 2 Report

Comments and Suggestions for Authors

The manuscript titled "Two C23-Steroids and a New Isocoumarin Metabolite from the Mangrove Sediment-Derived Fungus Penicillium sp. SCSIO 341429" investigated the pancreatic lipase (PL) and antioxidant inhibition activities of compounds isolated from the mangrove-sediment fungus Penicillium sp. Three new compounds and 12 known compounds were isolated, and their structures elucidated.

The study is well-supported with several figures illustrating the obtained results of all the assays. The supplementary file included the 1D and 2D NMR spectra of the new compounds. The discussion has provided a logical interpretation of the spectra with reliable identification of the newly identified compounds. In addition, the researchers conducted a comprehensive study to determine the absolute configurations of the new compounds through experimental and calculated ECD studies.

The attached PDF file of the manuscript contains comments for your consideration. Additionally, please find below some specific comments:

-       Some NMR correlations mentioned in the discussion are not found or clear in the supplementary spectra.

-       The discussion part for the MD studies needs to be improved.

-       In line 185, the sentence: “The result of the molecular docking of compound 15 was reported in previous studies” is not sufficient. Please mention the results of previous studies and compare them with the results obtained in the current study.

-       Also, there should be a comment on the variations between the In vitro inhibition assays of compounds 2, 14, and 15 against PL, with IC50 values of 32.77, 5.15, and 2.42 μM, and the results of MD showing very similar binding energies that are not consistent with the In vitro results.

-       I recommend a thorough grammar and language editing of the entire manuscript by a professional editing service.

Comments on the Quality of English Language

-       I recommend a thorough grammar and language editing of the entire manuscript by a professional editing service.

Author Response

Please see tha attachment.

Reviewer 3 Report

Comments and Suggestions for Authors

The review manuscript entitled “Two C23-Steroids and a New Isocoumarin Metabolite from 2 Mangrove Sediment-Derived Fungus Penicillium sp. SCSIO 3 41429” written by Haung and co-workers describes on isolation of two new C23 steroids and another new isocoumarine.

In spite of interesting structure of the new steroids, there are many serious problems on structural determination. Then the reviewer cannot recommend this paper to be published in marine drugs. The reasons for reject are listed below.

1.      Assignment of 1H-NMR listed in Table 1 is not worth believing. In the spectra seen  in Supporting information, many signals are overlapped. And the Table includes unreasonable data. What is coupling pattern noted in p for 11a, 12a of compound 1 and 5 of compound 2? There are many coupling constants over 20 Hz. Are they usual? 11a and 12a of coumpound 1 are almost same expect for the chemical shift value. Coupling constants do not agree between coupled protons. Why two protons at C-18 appeared in one signal. They are clearly not equivalent, and why are they doublet? There are so many.

The data do not agree with those of similar compound in reference 15.

2.      Both ECD spectra are not similar to experimentally obtained CD spectra for compound 1. Then, the authors cannot determine the configuration. The absolute configuration of compound 10 also cannot be determined from the same reason seen in Figure 5.

3.      The authors claim that H-3 and H-4 had trans configuration from the NOESY analysis. But they do not present the NOESY spectrum in SI. And it is curious how did they determine trans configuration from NOESY information. NOESY tells us how two protons are near. Furthermore, H-3 and H-4 of compound 10 do not coupled each other because they are simply coupled only with the connected methyl groups from data in Table 2. Are they aligned almost perpendicular to be J = 0 Hz? The COSY spectrum of compound 10 in SI has cross peaks between H-3 and H-4.

Above are enough evidences for reject.

Round 2

Reviewer 3 Report

Comments and Suggestions for Authors

The reviewer still suggests this paper to be rejected.

Revised Table 1 seems to be unsatisfied. The authors do not provide correct data. For example, chemical shift of 15b in compound 1 was revised to 1.58 ppm from 1.71-1.80 ppm in the original manuscript. H-16s in compound 1 are similarly doubtful. From the NMR spectrum of 1, the reviewer could not read H-3 as tt, it seems to be broad triplet. In contrary, H-5 at 2.69 ppm looks to be br d, and H-9 at 2.86 ppm as dd. Did the authors carefully analyzed NMRs? Similar as compound 2. Are singlet signals in compounds 1 and 2 seen lower field than 3.12 ppm methanol? HSQCs of them have the corresponding cross peaks.

 On compound 10, The authors did not respond to the reviewer’s question. Are H-3 and H-4 arranged to be perpendicular? Figures and the legends seen in different pages made the reviewer confused. Observation of the NOESY cross H-3/H-4 peak in compound 10  suggested they are near. Does it mean they are cis? Calculated ECD did not agree with the experimental one from that reason, the reviewer supposes. The authors claim they are trans in spite of the NOESY peak.